# In Silico Tools to Extract the Drug Design Information Content of Degradation Data: The Case of PROTACs Targeting the Androgen Receptor

**DOI:** 10.3390/molecules28031206

**Published:** 2023-01-26

**Authors:** Giulia Apprato, Giulia D’Agostini, Paolo Rossetti, Giuseppe Ermondi, Giulia Caron

**Affiliations:** Molecular Biotechnology and Health Sciences Department, University of Torino, Via Quarello, 15, 10135 Torino, Italy

**Keywords:** 2D molecular descriptors, androgen receptor, AR degraders, classification model, degradation, Degradation Cliff analysis, machine learning, permeability, PROTAC, ternary complex

## Abstract

Proteolysis-Targeting Chimeras (PROTACs) have recently emerged as a promising technology in the drug discovery landscape. Large interest in the degradation of the androgen receptor (AR) as a new anti-prostatic cancer strategy has resulted in several papers focusing on PROTACs against AR. This study explores the potential of a few in silico tools to extract drug design information from AR degradation data in the format often reported in the literature. After setting up a dataset of 92 PROTACs with consistent AR degradation values, we employed the Bemis–Murcko method for their classification. The resulting clusters were not informative in terms of structure–degradation relationship. Subsequently, we performed Degradation Cliff analysis and identified some key aspects conferring a positive contribution to activity, as well as some methodological limits when applying this approach to PROTACs. Linker structure degradation relationships were also investigated. Then, we built and characterized ternary complexes to validate previous results. Finally, we implemented machine learning classification models and showed that AR degradation for VHL-based but not CRBN-based PROTACs can be predicted from simple permeability-related 2D molecular descriptors.

## 1. Introduction

Proteolysis-Targeting Chimeras (PROTACs) are heterobifunctional molecules capable of inducing protein-targeted degradation via the recruitment of the ubiquitin–proteosome system (UPS) [1,2] PROTACs are made of three moieties: a warhead or protein of interest (POI) ligand, an E3 ubiquitin (E3) ligase-recruiting ligand and a linker connecting the two components. The ability of PROTACs to exploit the primary intracellular degradation pathway in eukaryotic cells derives from the simultaneous engaging of the E3 ligase and the POI. This results in the formation of a stable ternary complex mediating the polyubiquitination of the POI, later degraded by the 26S proteasome.

PROTACs (also referred to as degraders) have emerged as a promising and innovative tool in the drug discovery landscape. In fact, their event-driven mode of action considerably increases the druggable portion of the proteome, overcoming one of the major limitations of small-molecule receptor inhibitors [2]. Unlike small-molecule inhibitors, degraders target proteins with no known active sites or deep binding pockets, including scaffold proteins, protein complexes and transcription factors [3,4]. Among the most remarkable advantages of degraders, the long-lasting efficacy and the sub-stoichiometric mode of action (referred to as protein knockdown) should be mentioned [3,5]. Moreover, degraders seem to be a good therapeutic approach to overcome acquired resistance in cancer treatment [1,3,6]. 

Because PROTACs are made of three different building blocks, they result in larger structures; thus, finding the optimal combination of pharmacodynamics and pharmacokinetics is quite arduous and time-consuming. Today, this balance is sought through huge synthetic efforts. In practice, libraries of PROTACs are created from the combination of different E3-recruiting elements, POI ligands and linkers and by varying their conjugation points, linker lengths, flexibility and composition. The resulting compounds are then screened for degradation activity and are eventually further optimized (Bond & Crews, 2021) [4]. Such drug discovery efforts have produced a large amount of degradation data in the last ten years, which are often under-investigated. Retrospective studies are thus expected to be extremely useful to improve rational design strategies, in turn further accelerating the overall process. However, few papers and no general guidelines along these lines have been reported so far.

Collections of industrial data are being produced, and some of them have been published. However, no endorsed method for PROTAC degradation assessment is present. A few techniques, such as global mass spectrometry and CRISPR-Cas9 endogenous tagging, have recently been developed [7,8]. In particular, NanoBiT^®^ Luciferase, a complementation system formed by the HiBiT bioluminescent peptide (11 aa long) and a larger subunit named LgBiT, meets the demand of studying the dynamic nature of PROTACs in live cells. Indeed, the CRISPR-Cas9 knock-in of the HiBiT target allows following target engagement, ternary complex formation and ubiquitination in real time [9,10]. The main limitations of HiBiT endogenous tagging are its highly time-consuming nature and the need of checking the impact of the HiBiT-tag on the protein structure and localization. Western Blotting (WB) remains the most widely employed assay to quantify POI degradation. The caveat of immunoblotting-based degradation data arises from the several limitations of the method. First, the overall process involves a complex series of steps influenced by subjective choices and user expertise, affecting reproducibility. Second, the availability of high-quality specific antibodies for target detection and the lack of adaptability for high throughput screening are other crucial limiting factors. Finally, WB cannot be performed in live cells, thus providing an endpoint after cell lysis. However, degradation is a highly dynamic process; thus, it requires monitoring at multiple time points [7,8]. To increase the reproducibility and reliability of the assay, Dmax, which is the maximum level of achievable degradation, and DC50 (half of the maximum degradation concentration) are the preferred parameters to be obtained. However, in the early drug discovery phase, a degradation percentage value assessed at a single concentration is often the only information available. 

Another issue affecting the degradation activity is cell permeability. Indeed, degradation can take place only if the degrader crosses cell membranes and enters the cell. In this context, the dBET1–dBET6 degrader pair is a paradigmatic case [11]. Those Bromodomain-containing protein 4 (BRD4)-targeting degraders are characterized by the same warhead and CRBN-recruiting ligand, but they differ in the length of the alkyl linker. Specifically, the linker of dBET1 is shorter and more rigid (four Carbon atoms), whereas dBET6 has a longer and more flexible linker (eight Carbon atoms). Although showing comparable binding profiles, dBET6 is two orders of magnitude more active than dBET1 (sub-nanomolar range activity). This different target degradation profile seems to be related to the increased cell permeability of dBET6, as demonstrated by cellular target engagement and Caco transwell assays [11].

ARV-110 (Figure 1) is an oral PROTAC that successfully reached phase II clinical trials (clinicaltrial.gov, NCT3888612). This PROTAC selectively targets the androgen receptor (AR). This latter is amplified or mutated in metastatic castration-resistant prostate cancer (mCRPC), a late form of prostate cancer that is currently incurable [12,13]. 

Considering the universal interest in AR degradation also led by the success of ARV-110, many recent studies have focused on targeting AR with PROTACs. For instance, three recently published papers by the Wang group [14,15,16] have reported several PROTACs targeting AR, together with their degradation data determined under coherent experimental conditions (cell lines, exposure time and concentrations). This pool of degradation data provides a consistent dataset that deserves further analysis, at least from our viewpoint. We are aware that the reported compounds are just the optimized derivatives resulting from a wider synthetic effort, and that the study of the entire library could provide more insights; however, again, this is the only data format at present that is available in the literature.

The main aim of this study was to apply computational approaches initially developed for small molecules to a PROTAC dataset. We aimed to extract its entire information content and investigate its potential application to the design of new degraders. Moreover, we highlighted the application field, the pros and the cons of each exploited method. In this context, many molecular modeling approaches may be used to evaluate the degradation potential of PROTACs, but their review is beyond the aim of the paper. Here, we selected and applied tools useable in early drug discovery. In practice, powerful but highly CPU-demanding strategies, such as molecular dynamics [17,18] and PROTAC-induced ubiquitination prediction [19], were not included in the study.

The PROTAC dataset was first structurally characterized through the Bemis–Murcko framework analysis combined with the Matched Molecular Pair approach. Then, the degradation profile of each cluster was analyzed. Next, we checked the applicability of the ligand-based Degradation Cliff (DC) [20] strategy and performed a linker analysis using a simple Structure Activity Relationship (SAR) approach on selected subsets of compounds. The DC results were validated with a structure-based approach by characterizing the ternary complexes of selected degraders [21]. Finally, we calculated a pool of previously described permeability-related 2D molecular descriptors [22] and obtained a machine learning (ML) classification model to discriminate the AR degradation capacity of the PROTACs from the dataset. 

Overall, this paper reports a first attempt to analyze the published PROTACs dataset to deconvolute information needed to drive the design of degrader candidates with a better drug-like profile.

## 2. Results and Discussion

### 2.1. Dataset Description and Preliminary Analysis

We merged three uniform and consistent datasets of AR degraders published by the same research group [14,15,16] to obtain a final dataset consisting of 92 compounds, including 53 VHL- and 39 CRBN-recruiting PROTACs (Appendix A). A notation specific for our study was adopted; compounds 2–22 from Wang et al. [14] were annotated as c01–c21, compounds 8–41 [16] were annotated as c22–c55, and the others [15] were annotated as c56–c92.

Degradation activity (Appendix A) was measured at two different concentrations: 0.1 μM and 1 μM (we used both sets of data, but in general, in the text, we refer to that of 0.1 μM) in prostate cancer cell lines LNCaP and VCaP, after, respectively, 6 h and 24 h of incubation for compounds c01–c55 and compounds c56–c92. As reported in the original papers, c01–c55 degradation activity was assessed in LNCaP, whereas c56–c92 was measured in VCaP. In both cell lines, fast kinetics of degradation was observed for the most active compounds. The Dmax and DC50 values reported covered the same order of magnitude. An exception observed by the authors regards compound c89 (original notation, 45), which seems to be selectively more potent in VCaP cells. Both cell lines are the most widely used models for castration-resistant prostate cancer. LNCaP carries a specific AR mutation (T878A), whereas VCaP is transformed by the amplification of wild-type AR. It is suggested that effective AR degraders designed against wild-type AR might fail in the presence of mutations, but the degradation activity correlates in the two cell lines. Therefore, we did not consider the isolated case of c89 as an issue. As discussed in the Introduction, the format of the degradation data is a degradation percentage. This is not the preferred format to set up a modeling study, but because it is representative of early drug discovery phases, we proceeded further by checking its usefulness.

Two different activity cutoffs, 50% (data not shown) and 75% (in the text), were considered. Notably, with this choice, compounds having a degradation lower than 75% were deemed as inactive. In Appendix A, the activity distribution of degraders at 0.1 μM is reported, with active degraders in green and inactive degraders in red. The activity distribution for CRBN- and VHL-recruiting degraders is also reported in Appendix A. A slight data imbalance toward inactive compounds is overall present, as well as in both subsets.

### 2.2. Dataset Structural Characterization and Clustering

The Bemis–Murcko fragmentation method [23], combined with a clustering procedure based on the Matched Molecular Pair (MMP) approach, and Similarity Analysis [24] are widely used computational methods to explore/characterize datasets of interest. Notably, these chemoinformatic tools have never been applied to PROTACs in published works. 

First, we focused on the entire PROTAC structure. The Murcko Scaffold Analysis (Appendix A) combined with the MMP strategy allowed us to group the PROTACs into 12 clusters, named a-n (Appendix A). Compounds both sharing the same scaffold and having a different scaffold but presenting a single localized structural feature (i.e., a ring system substitution) were clustered together. The Similarity Analysis was performed on the dataset as well; it exploited SkelSpheres, molecular descriptors implemented in DataWarrior, to identify molecular fingerprints, as well as Tanimoto’s equation to quantify compound similarity (Appendix A) [25]. Notably, SkelSpheres were not developed for large molecules, such as PROTACs (the MW of the considered compounds ranges from 680 to 1155 Da). However, the clusters found through the Similarity Analysis were coherent and quite superposable with the ones identified through the Murcko/MMP approach.

Figure 2A shows a boxplot in which each cluster (the x-axis reports the cluster names) is colored according to the degradation activity (assessed at 0.1 μM). Despite the structural similarity of the compounds included within the same cluster, the degradation activity is not homogeneous; clusters b, c and m are prevalently characterized by highly active compounds, whereas the others display heterogeneous degradation activity.

The Bemis–Murcko framework analysis was also performed on the three building blocks separately. Four different PROTAC clusters were identified through the Matched Molecular Pairs strategy considering the degrader warhead (Figure 2B; Appendix A). The presence of a specific warhead framework does not correlate with a particularly active or inactive cluster of compounds, which is coherent with the known concept that PROTAC activity is a complex phenomenon. 

Figure 2C shows the activity distribution of the degraders grouped according to the E3 ligand. Three clusters were found and named CRBN, VHL-a and VHL-b (Appendix A). In the CRBN cluster, the activity distribution is only related to modifications of the warhead and the linker. Notably, the VHL-a cluster is characterized by a high number of inactive compounds, whereas the VHL-b class is populated by active PROTACs, except for c07, c21 and c45.

Figure 2D shows the activity distribution of the degraders grouped according to the linker structure (Appendix A). The linker classification based on the Bemis–Murcko framework analysis appears to be biased by the linker optimization strategy followed by the authors during synthesis. Thus, different linker lengths and compositions were examined for both CRBN- and VHL-recruiting degraders, which led to a heterogeneous set of tested structures. Considering that the C-c cluster includes ARV-110-derived linkers, it is not surprising that mainly active compounds populate this cluster.

Overall, the analysis of the degradation activity based on Bemis–Murcko frameworks clustered via Matched Molecular Pair does not provide relevant information to be applied in the design of new AR degraders. This finding confirms that PROTAC degradation is not simply the sum of the activity of each building block; rather, it is the result of a synergistic effect.

In light of these considerations, we moved on to the hypothesis that small modifications are expected to modulate PROTAC degradation activity.

### 2.3. Degradation Cliffs

Activity cliffs (ACs) are generally defined as pairs or groups of structurally similar compounds that are active against the same target and characterized by a substantial potency difference [20,25]. The analysis of ACs is relevant in the early stages of the optimization of compounds when the potency needs to be improved. This approach is commonly applied to small-molecule inhibitors. We reasoned that we could try to identify Degradation Cliffs (DCs), i.e., structurally similar PROTACs degrading the same target but displaying large differences in degradation. 

The output from the DC analysis consisted of 431 pairs. Nevertheless, only 15 of them satisfied the Delta Activity threshold at both considered concentrations (Appendix A and Methods). In Figure 3, we report three representative pairs presenting modifications of the warhead (examples of E3 ligand-related DC pairs are reported in Appendix A).

The c71–c77 pair suggests that the inclusion of the amidic nitrogen in the cycle is detrimental for degradation, and the c71–c74 and c71–c76 couples highlight the importance of the ring size. Indeed, the five-membered ring seems to be insufficiently bulky for stabilizing warhead–AR interactions, whereas bulkier moieties such as the cyclo-octyl ring of c76 cause a sharp decrease in activity. It should be pointed out that only the first pair (c71–c77) was identified through the Delta-Activity-based Degradation Cliff method. The remaining pairs (c71–c74 and c71–c76) were not detected through the DC approach and were manually selected. This methodological issue can be due to several reasons, such as the similarity definition and the chosen descriptor, the Delta Activity method per se and/or the chosen cut-off. Moreover, the loss of information can be also attributed to the quality of the degradation data; low-quality or inconsistent data, such as single-point degradation, may affect the accuracy. 

Overall, Degradation Cliff analysis is a powerful and useful tool to identify key structural modifications impacting degraders’ activity, but consistent degradation data and new PROTAC-tailored similarity descriptors are needed to take the most out of this technique when applied to large and flexible structures.

### 2.4. Linker–Degradation Relationships (LDRs)

Linker modification (i.e., length, flexibility and atomic composition) is the most interesting opportunity for both discovering hits and optimizing PROTAC leads [17,26,27,28]. Although linkers do not have a direct role in degradation (although sometimes they establish nonspecific interactions with POI and E3 ligase), they heavily affect PROTACs’ future as drug candidates by allowing the binding of the degrader with their POI and E3 ligase and by impacting the pharmacokinetic profile as well.

Linker–degradation relationships (LDRs) are not so common in the published literature on degraders due to the lack of a suitable series, i.e., PROTACs with the same E3 ligase ligand, POI ligand, linkage point on each ligand, linkage functionality (such as an amide bond or aniline linkages) and linker chemical composition (e.g., aliphatic or PEG). The investigated dataset includes two series of PROTACs for which LDRs deserve to be investigated (representative chemical structures in Appendix A). Figure 4 shows how degradation (0.1 μM) varies with the linker length. Although both subsets show a somewhat rough parabolic trend, the optimal linker length varies with the series, being shorter for CRBN-PROTACs than for VHL derivatives.

Linear alkyl (and ether) linkers are increasingly being replaced by motifs able to impart some molecular rigidity, such as heterocyclic scaffolds and alkynes [27]. The investigated dataset contains a third series of comparable (see above) rigid derivatives (CRBN c83–c89). This subset includes seven PROTACs with four types of rigid linkers (i.e., piperazine-methylene-piperidine, piperazine-piperidine, piperazine-methylene-azetidine and azetidine-piperazine, Appendix A). Notably, no correspondent flexible series can be individuated. The degradation data of c83–c89 range from 86 to 99 and show that, at least in the case of AR receptors, rigid, protonated linkers sharing a similar optimized length always provide an excellent degradation value almost independently from their structure.

### 2.5. Ternary Complexes

As previously discussed, POI ubiquitination and the subsequent degradation are dependent on the formation of the ternary complex (TC). A PROTAC engages both the target protein and the E3 ligase component simultaneously to form the TC. Therefore, the previously identified ligand-based couples were confirmed to adopt a structure-based strategy based on TC modeling.

Figure 5 shows the ternary complex of ARV-110, highlighting the main intermolecular interactions governing the binding of the PROTAC with AR (depicted in yellow) and CRBN (depicted in light violet).

ARV-110 was used as a reference compound to identify the key interactions involved in the ternary complex formation and the relative binding energy (Figure 5B,C). These data were then compared with those obtained from the TC resulting from the replacement of ARV-110 with the two PROTACs of a DC pair, i.e., c71–c77 (Figure 6, see below for more details about this pair). Figure 6 highlights that, whereas the ternary complex of c71 mostly preserves the interactions of ARV-110, c77 does not. The most important difference is the loss of the two HBs formed by the cyano moiety of the warhead (Gln711 and Arg752), which is fundamental for binding the POI (Appendix A). Moreover, the disruption of several interactions between the E3 ligand and the CRBN receptor is observed because of the instability of the warhead pose. The overall stability of the ternary complex strongly suffers from these losses, as it appears clear considering the interaction energy (Figure 6B,C). 

To also focus on VHL-recruiting degraders, we investigated the ternary complexes related to the DC pair c48–c52 that present a crucial modification in the E3 ligand moiety. These two compounds are characterized by a different stereochemistry of the carbon atom bound to the phenyl group. c48 (original notation, ARD-69) is the most active degrader, as published by Han et al. [16], and it has been considered the reference compound to identify the most significant interactions present in the ternary complex. As shown in Appendix A, c52 (the inactive degrader of the pair) nearly loses all interactions with the VHL receptor, although it maintains those between the warhead and POI (same distance and energy). As a result, the overall interaction energy harshly drops to half of the value of c48 TC. This confirms the crucial role of stereochemistry in the E3 ligand moiety.

Overall, the ternary complex modeling proved to be a robust method to validate the Degradation Cliff results. Although we recognize that this low throughput in silico tool cannot be employed as an early screening default technique, it could provide useful insights in understanding the gain or loss of activity of a more restricted pool of degraders.

### 2.6. Molecular Properties Analysis and Classification Models

As discussed in the introduction, degradation data implicitly include permeability contributions because PROTACs must first enter the cell. To better understand this aspect, we characterized the dataset focusing on a restricted but representative pool of seven 2D permeability-related molecular descriptors of polarity, size/hydrophobicity and flexibility, obviously related to permeability [22]. Notably, in a previous work, we reasoned that, to date, the application of 3D descriptors to large and flexible structures such as PROTACs seems premature due to the issues associated with the conformer selection [29]. The three chosen polar descriptors are the topological polar surface area (TPSA), intended as a measure of the maximum polarity that a degrader can express, the number of hydrogen bond donors (HBD or nHDon) and the hydrogen bond acceptors (HBA or nHAcc) atom counts. Molecular weight (MW) was chosen as a size descriptor, and the number of carbon (nC), and the number of aromatic rings (nAR) were chosen as hydrophobicity-related descriptors. Finally, because of PROTACs’ flexible beyond rule of five molecules (bRo5), we chose Kier’s flexibility index (PHI) as a flexibility descriptor. As explained in our previous work, log P was not considered because, to date, no log P calculator has been able to properly predict bRo5 compounds and thus PROTACs as well. 

The seven descriptors were calculated for all 92 compounds. They are not normally distributed throughout the dataset (Appendix A). The correlation matrix of the molecular descriptors is reported in Appendix A. As expected, TPSA correlates with nHAcc and nHDon, as well as with non-polar descriptors including nAR, MW and the flexibility descriptor PHI. 

First, we exploited simple but highly informative infographic tools. Figure 7 shows the TPSA variation in the CRBN (Figure 7A) and the VHL (Figure 7B) subclasses. Bars are colored in green (active) and red (inactive) using a threshold of 75% degradation activity (0.1 μM). Interestingly, TPSA seems to play a role in separating AR PROTACs according to their degradation profiles, and this appears to be true especially for VHL degraders. 

This finding suggests setting up a machine learning (ML) strategy looking for a classification model based on descriptors related to permeability, considering VHL and CRBN as two separate classes. We randomly split each subset into training and test sets (see Methods) and trained the binary classification models with the following input: (i) VHL subclass, (ii) CRBN subclass and (iii) the overall dataset. To this aim, five different classifiers were selected: two tree-based algorithms, Random Forest (RF) and Random Tree (RT); one probability-based classifier, naïve Bayes; one distance-based classifier, K-nearest neighbors (KNN); and a Support Vector Machine (SVM) algorithm with a linear kernel [30,31,32,33,34,35]. Each model was trained using 10-fold cross-validation on the training set [36]. The test set was then used for external validation to evaluate the predictive power and the strength of the model (see Materials and Methods for more details). Each model was evaluated in terms of sensitivity or True Positive Rate (TPR), specificity or True Negative Rate (TNR), Receiver Operating Curve (ROC) Area and Matthews Correlation Coefficient (MCC) [37]. In an ideal model, the ROC Area is near 1 [38], whereas an MCC value larger than 0.4 is the threshold to identify the presence of a robust model [37,39].

The performance statistics of the best-performing models (activity measured at a degrader concentration of 0.1 μM) are in Table 1 (all the models implemented with activity at 0.1 and 1 μM are reported in Appendix A) [35,36,37]. The different machine learning classification tools provided similar results; however, we decided to focus on RF models because they showed good performances on both the training and the test sets. This is particularly relevant for avoiding the risk of overfitting, a risk that is present due to the small dimensions of the dataset. Notably, this is in line with the literature, which supports that RF methods are relatively robust against overfitting [40].

Table 1 shows that the VHL models perform well, whereas CRBN models do not seem to perform likewise. Indeed, whereas VHL models can correctly classify 35 out of 42 compounds (83.3%), CRBN identifies only 18 out of 31 (58.1%) active compounds. The ROC Area and the MCC values further confirm the goodness of VHL models. Y-randomization, in which the original degradation activity values of training sets were replaced by randomly generated numbers prior to model training, also supported our findings (Appendix A). Considering that CRBN degraders are about 40% of the overall number of compounds, it is not completely unexpected that classification does not properly perform on the entire dataset.

A further analysis revealed that nHAcc and TPSA are first-rank attributes with the greatest weight on the implementation of the VHL 0.1 μM activity model (see Materials and Methods for details). This is confirmed by the three descriptors’ (nHAcc, TPSA and MW, Appendix A) model performance statistics, which are of lower quality than those reported in Table 1 but which are good enough to confirm that the models are not affected by overfitting issues. 

Overall, we showed that the degradation of VHL- but not CRBN-PROTACs can be predicted from a classification model based on 2D descriptors related to permeability.

## 3. Materials and Methods

### 3.1. Dataset Creation

PROTACs SMILES and activity data were retrieved from the original papers. Compounds were divided according to the E3 ligase recruited for degradation, and each group, VHL and CRBN, were further randomly split using R Studio 2021.09.2 build 382 (R software version 4.1) into training and test sets, respecting the 80:20 ratio. The entire dataset was randomly split into training (74 compounds) and test (18 compounds) sets using R software as well.

### 3.2. Bemis–Murcko Framework Analysis and Murcko Molecular Cluster Creation

The Bemis–Murcko framework analysis was performed using OSIRIS Datawarrior 5.5.0 (https://openmolecules.org/datawarrior/index.html (accessed on 7 June 2022)). The software generates two types of frameworks: the Murcko Skeleton and the Murcko Scaffold. The Murcko skeleton represents the molecular core structure resulting after the removal of the side-chain atoms and atom labels, focusing only on the graph properties of the molecule. The Murcko scaffold contains all plain ring systems of the given molecule plus all direct connections between them, maintaining the atom labels and stereochemistry, which provides the atomic properties of the given molecule [41]. 

Murcko Molecular Clusters were obtained by considering the Murcko Scaffolds identified for the compounds included in the dataset and by adopting the Matched Molecular Pair strategy to collect different Scaffolds. Compounds sharing the same scaffold were grouped in the same cluster, also including compounds differing in a single localized structural feature (i.e., ring system substitution).

### 3.3. Degradation Cliff Analysis

Degradation Cliff (DC) Analysis was performed using OSIRIS Datawarrior 5.5.0 (https://openmolecules.org/datawarrior/index.html (accessed on 10 June 2022)). The following were used as input: a .csv file containing molecule identifiers, the SMILES strings of molecules and the degradation activity values at 0.1 uM and 1 uM (the analysis was carried out using both concentration values separately). The procedure to identify DC is schematized in Appendix A. The SkelSpheres descriptor, Tanimoto’s index (similarity threshold set to 90%) and the Delta activity parameter were calculated as implemented in Datawarrior. 

### 3.4. Ternary Complexes

Ternary complexes were modeled using docking and minimization steps with the linker as the geometrical constraint to find the reciprocal position of the two proteins. Structures of POI and E3 ligases co-crystallized with molecules identical or closely related to warheads and E3 binders (Appendix A) were used. 

The AR W741L -Bicalutamide complex, completely similar to WT-AR, was used to suitably orient the pocket’s side chains for NSAA binding. The mutation was reverted with the MOE protein builder tool (https://www.chemcomp.com/Products.htm (accessed on 22 September 2022)), and nearby residues were minimized. The warheads were docked in AR using SwissDock (http://www.swissdock.ch/ (accessed on 22 September 2022)) by defining a cube of 8x8x8 in the DHT-NSAAs pocket. The obtained poses were visualized with Chimera (https://www.cgl.ucsf.edu/chimera/ (accessed on 26 September 2022)), and the best ΔG pose was retained as the best pose. The E3L–binder complexes were downloaded from the RCSB PDB website (https://www.rcsb.org/ (accessed on 22 September 2022)). The linker was built on both the E3L and warhead sides using VEGA-ZZ (https://www.ddl.unimi.it/cms/index.php?Software_projects:VEGA_ZZ (accessed on 26 September 2022)). The complexes, including the E3L–binder complex and the linker, as well as the one with the POI–warhead and the linker, were both minimized in MOE using the MMFF94x force field (default settings). The system was protonated at a pH of 7.4 and was then minimized. The minimization was the steepest descent gradient minimization, with an RMS of 0.05 Kcal/mol/A^2^, and it was performed considering planar systems as rigid bodies. Initially, the minimization protocol was applied only to Ligand atoms and the protein atoms in a radius of 4.5 A, aiming to define the linker’s vector. Successively, the two complexes were opened in Chimera, and the Model Panel tool was used to reciprocally move their coordinates, superposing the vector designed by the two linkers. The superposition between linkers must correspond to a protein pose that should not cause extensive clashes. Making compromises between clash formation and respecting linker vectors, a rough model of the ternary complex was obtained. This ternary complex was finally built using MOE by first deleting one of the two linkers and creating the bond on the other. Protonation was conducted as before. The first refinement was devoted to solving protein–protein clashes; using the Protein builder tool, it was possible to select the residues involved in clashes (individuated with the Structure preparation tool) and to solve them using a minimization protocol dedicated to proteins. Then, the first minimization step involving the PROTAC and the neighboring (4.5A radius) atoms was performed to have realistic bond lengths in the PROTAC. Then, in the second minimization step, all atoms were left free to move. Finally, the structure preparation tool was newly checked for eventual residual clashes that needed to be solved.

### 3.5. Molecular Descriptors

A single .txt file with PROTACs SMILES and IDs was submitted to the licensed software AlvaDesc (Alvascience, version 1.0.18, n 2020, http://www.alvascience.com/alvadesc/ (accessed on 4 October 2022)) for MW, nC, PHI, nHAcc, nHDon and TPSA calculations. A single .csv file with SMILES and degrader IDs was submitted to OSIRIS DataWarrior (version 5.5.0, http://www.openmolecules.org/datawarrior/ (accessed on 4 October 2022)) for nAR prediction. Predicted data were collected into a .csv file containing activity degradation data, and they were further analyzed. 

### 3.6. Data Visualization

Pearson correlation matrices (SI) were obtained with R using corrplot (Pearson correlation plot), ggplot2, ggalt and devtools. The other infographics, including molecular descriptor distributions and activity boxplots, were created using DataWarrior.

### 3.7. Classification Models

Classification models were created with Waikato University software WEKA version 3.8.6 (https://www.cs.waikato.ac.nz/ml/weka/ (accessed on 4 October 2022)). A .csv file containing molecular descriptors and activity at either a 0.1 uM or 1 uM concentration was uploaded in the WEKA Explorer Environment. The VHL and CRBN datasets were considered either separately or together, and the results are reported for both cases. The threshold was chosen arbitrarily, and it was set to 75% degradation activity, with compounds having a score >= 75 being considered active and those with a score < 75 being considered inactive. Ten-fold cross validation was chosen as the experiment type to train the models, and Random Forest, Random Tree, naïve Bayes and IBk with k = 5 (KNN) were set as classifier algorithms. The classification was performed using default values for each algorithm, except for IBk. Several K values were tested, and the final model was built by setting K = 5. The obtained models were then tested on the external test set to evaluate their performance via the external validation procedure. Sensitivity, specificity, ROC Area and MCC were used to evaluate each model. An MCC higher than 0.4 in the training set was set as the cut-off to identify well-performing models. 

To define the descriptors impacting the VHL classification model the most, WEKA attribute evaluation was performed. InfoGainAttributEval, CorrelationAttributEval, CsfSubsetEval and OneRAttributEval were applied (Table 2). InfoGainAttributEval evaluates the worth of an attribute by measuring the information gain with respect to the class. It is a probability-based tool that considers the probability that a specific Attribute is associated with a specific Class, as reported in Equation (1):(1)InfoGain(Class, Attribute)=H(Class)−H(Class | Attribute)

CorrelationAttributEval instead evaluates the correlation via the Pearson’s correlation coefficient between the single Attribute and the Class. CsfSubsetEval evaluates the worth of each Attribute by considering its individual ability to correctly predict the Class; the chosen features are highly correlated with the Class and have a low intercorrelation with each other. OneRAttributEval evaluates the worth of each Attribute according to the OneR classifier. For more details, refer to WEKA software version 3.8.6. 

A Support Vector Machine (SVM) implemented in Orange version 3.34 (https://orangedatamining.com/ (accessed on 10 January 2023)) was also used to obtain classification models. SVMs are widely successful machine learning tools developed for two-class classification problems [34,35]. Due to the relatively small amount of data, we preferred to use a linear kernel to avoid overfitting risks.

## 4. Conclusions

The growing interest in PROTACs is producing a large amount of data that are generally poorly investigated for their entire information content on drug design. This work represents a first attempt to analyze the literature’s PROTAC dataset with an unconventional strategy based on a few in silico tools, generally not used in combination. 

While exploring the PROTAC literature, we realized that degradation data are quite heterogeneous, especially concerning experimental set up, principally including degrader concentrations, incubation times and cell lines. Dmax and DC50, which are gradually becoming the gold standard for PROTAC degradation assessment, are crucial to perform accurate SAR analysis. Unfortunately, in the first steps of the drug discovery process, data received from assessing the degradation at two or three different concentrations are the only available data. Therefore, we had to face the evidence that no default computational approach is currently available to support a rational PROTAC design based on modest-quality degradation data. Moreover, the standard assay used to assess target degradation remains Western Blotting, despite its low reproducibility and semi-quantitative measurements, although recent techniques allow a more accurate and real-time target degradation assessment (e.g., NanoBit^®^ Luciferase combined with CRISPR-Cas9 technology).

The structural analysis performed via Murcko frameworks and the identification of Matched Molecular Pair clusters did not show any issue when applied to the complex degrader structure. However, the framework analysis produced no significant result in terms of structure–degradation relationship, and it supports the idea that the prediction of degradation from PROTAC building blocks is not recommended. We then focused on single-point modifications and performed an Activity Cliff analysis, here called Degradation Cliff (DC). This produced a good although incomplete amount of information in the identification of the structural moieties responsible for consistent AR degradation differences. The implementation of new indexes and similarity descriptors specifically designed for large and flexible PROTAC structures is expected to improve the DC performance.

Linker–degradation relationships revealed that VHL and CRBN AR PROTACs need different linker lengths to optimize their degradation skills, and rigid linkers with an optimal length can degrade the target independently from their structure. 

Because of the ligand-based nature of the DC approach, we compared DC findings with a structure-based method, i.e., ternary complex characterization, which supported the previous results. The combination of the two strategies is strongly recommended, if possible, to avoid bias due to the use of single techniques. 

Finally, we used machine learning techniques to build classification models that may be used as virtual filters in early drug discovery projects. Two-dimensional-based classification models discriminated between VHL-based PROTACs that produce low-medium and high AR degradation, but they failed with CRBN derivatives. The classification models highlighted the potential and often underestimated contribution of permeability to degradation data. Polarity descriptors, specifically TPSA, seem to play a pivotal role in the identification of potent AR VHL-PROTAC degraders. A possible explanation of the absence of classification models for CRBN-PROTACs may be related to the synthetic strategy applied during the design of the selected compounds. Indeed, CRBN-PROTACs are characterized by single localized modifications that do not significantly alter degraders’ molecular properties, such as TPSA. 

Finally, the transfer of these results to other degraders currently does not seem feasible. However, we hope that this work will concretely help with degradation data analysis and encourage the production and sharing of high-quality degradation data. 

## Figures and Tables

**Figure 1 molecules-28-01206-f001:**
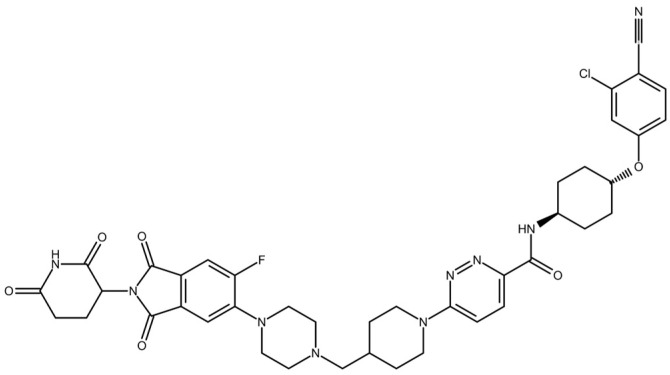
ARV-110.

**Figure 2 molecules-28-01206-f002:**
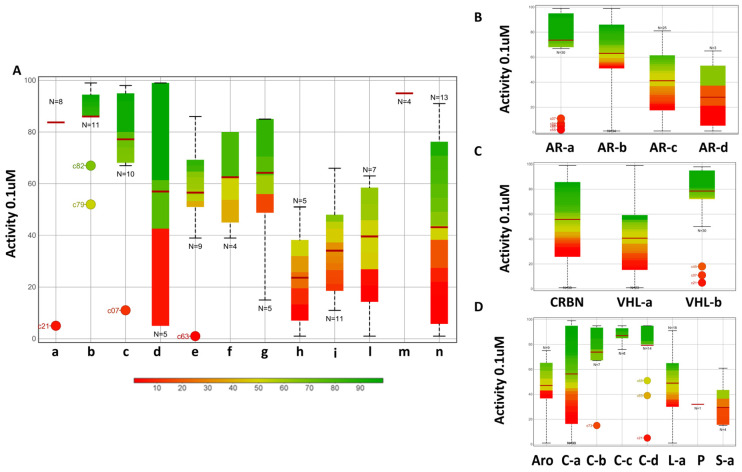
Boxplots reporting the activity distribution per cluster identified through the Bemis–Murcko Molecular Cluster analysis of different building blocks. On the x-axis, the cluster name is reported, and on the y-axis, the activity assessed at 0.1 uM is shown. Boxplots are colored according to the activity of degraders included within the cluster. Moreover, the number of compounds present within the cluster is reported as well, and the average activity value is represented with a red line. (**A**) Entire PROTAC structure. (**B**) Warhead. (**C**) E3 ligand. (**D**) Linker.

**Figure 3 molecules-28-01206-f003:**
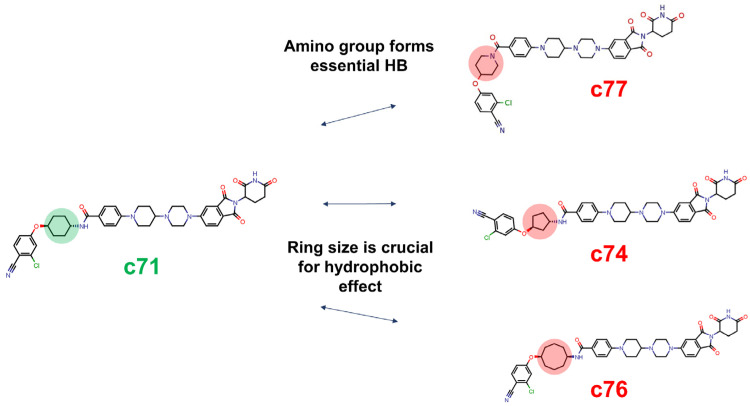
Warhead-related Degradation Cliffs. The arrows connect compounds included in the same pair, and the features responsible for the DC pair are highlighted in green (active-compound-related modification) or red (inactive-compound-related modification).

**Figure 4 molecules-28-01206-f004:**
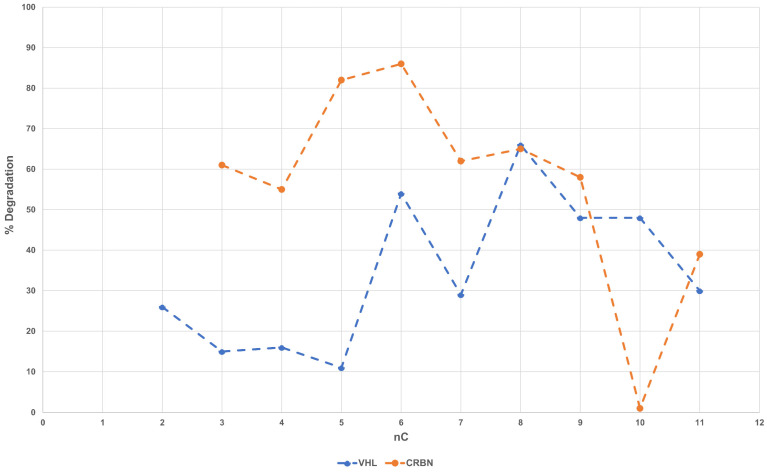
Degradation 0.1 μM vs. number of carbon atoms in the linker.

**Figure 5 molecules-28-01206-f005:**
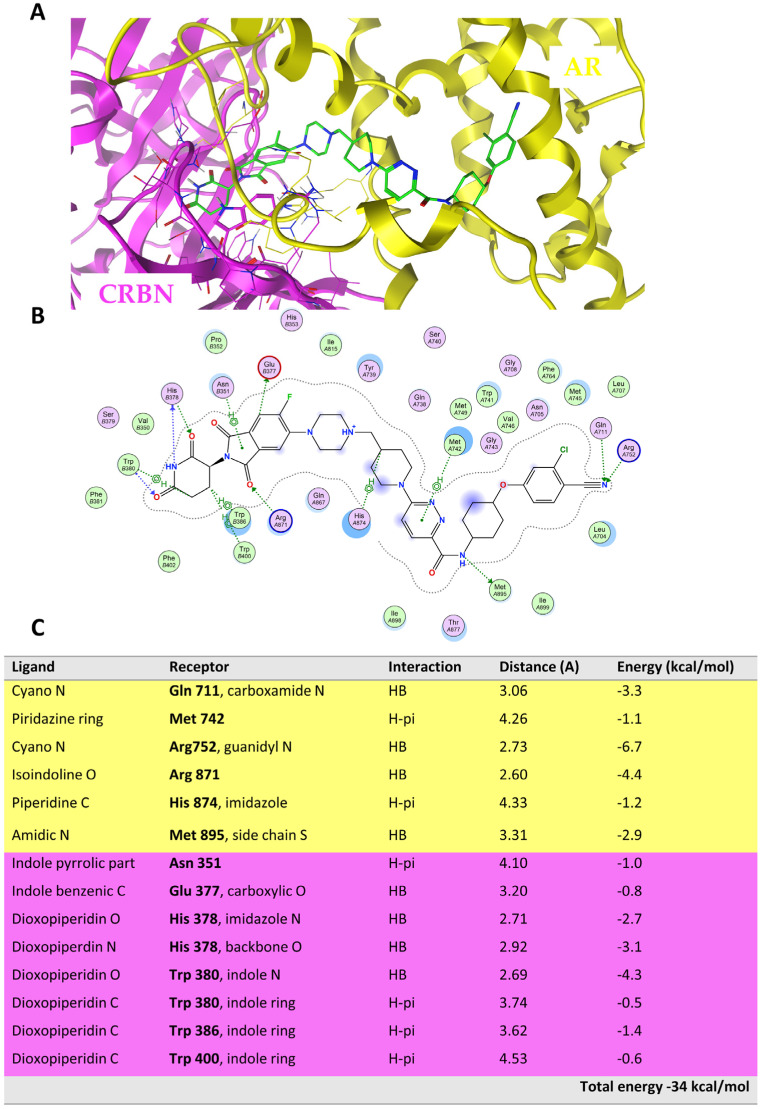
(**A**) The ternary complex is shown. ARV-110 is shown in green, AR is shown in yellow, and CRBN is shown in light violet. (**B**) Ligand interaction chart. (**C**) Interaction table highlighting the main interactions between ARV-110 AR (yellow) and CRBN (light violet).

**Figure 6 molecules-28-01206-f006:**
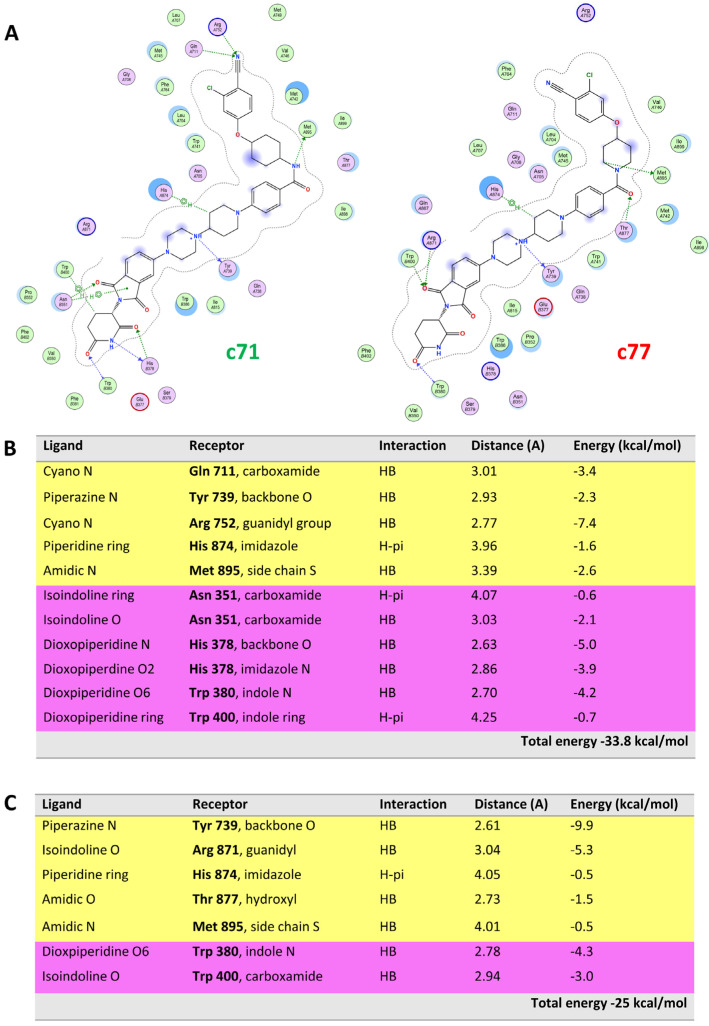
Comparison of c71 (active degrader) and c77 (inactive degrader). (**A**) Ligand interaction chart of c71 and c77. (**B**) c71 table of interactions with AR (yellow) and CRBN (light violet) are reported. (**C**) c77 table of interactions with AR (yellow) and CRBN (light violet).

**Figure 7 molecules-28-01206-f007:**
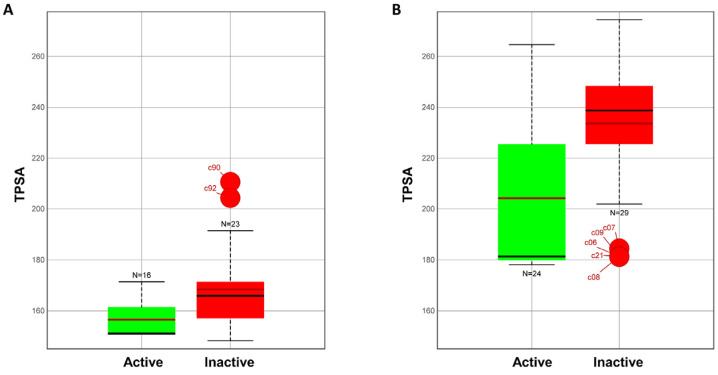
TPSA (polarity) distribution: (**A**) CRBN subclass. (**B**) VHL subclass.

**Table 1 molecules-28-01206-t001:** Performance metrics for the best-performing classification models, with 75% degradation activity as active/non-active threshold for 0.1 μM activity. Sensitivity refers to the ability of the model to correctly identify active compounds, and specificity refers to the capacity of discriminating non-active compounds from false negative ones. MCC is the Matthews Correlation Coefficient, and it represents how well the model can discriminate between the classes, active and non-active compounds. The ROC curve is a measure of sensitivity and specificity over a wide range of all possible threshold values [37]. RF: Random Forest classifier; RT: Random Tree classifier; VHL: Von Hippel-Lindau protein; CRBN: Cereblon protein.

Model		Sensitivity (TPR)	Specificity (TNR)	MCC	ROC Area
Entire dataset	RF (training set)	0.781	0.690	0.468	0.770
	RF (test set)	0.750	0.700	0.447	0.813
VHL	RF (training set)	0.737	0.913	0.666	0.863
	RF (test set)	0.800	0.833	0.633	0.867
	RT (training set)	0.737	0.870	0.615	0.819
	RT (test set)	0.800	0.833	0.633	0.817
CRBN	RF (training set)	0.556	0.615	0.169	0.615
	RF (test set)	0.800	1	0.775	0.967
	RT (training set)	0.444	0.692	0.139	0.568
	RT (test set)	1	0.667	0.745	0.833

**Table 2 molecules-28-01206-t002:** Attribute evaluators and first-ranked attributes for each attribute.

Attribute Evaluator	First-Ranked Attributes
InfoGainAttributEval	nHAcc, TPSA
CorrelationAttributEval	nHAcc, TPSA
CfsSubsetEval	nHAcc
OneRAttributEval	nHAcc, nHDon, TPSA

## Data Availability

The data presented in this study are available on request from the corresponding author.

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
