# Peer review of "In Silico Tools to Extract the Drug Design Information Content of Degradation Data: The Case of PROTACs Targeting the Androgen Receptor"

_molecules, 2023, doi:10.3390/molecules28031206_

Round 1
Reviewer 1 Report
In order to extract information that can inform the design of PROTACs, the authors describe a thorough study of the androgen receptor degradation data in this paper. The study covers a broad range of the properties of the degraders or the complex they form with proteins. The work is solid, and the authors make clear arguments regarding the choice of methods and provide proper discussions on the results.
Although it seems sense to use machine learning approaches in this study because they have proven to be effective at sifting out key features from challenging datasets, the choice of machine learning method needs more justification. Because the number of dataset is relatively small in this study, such a limitation could lead to the failure of many machine learning methods. With this concern, I would suggest the following point might need to be addressed or discussed:
- Could the author elaborate more on their decision-making process for ML methods? How can they ensure that the overfitting issue won't arise?
- Will simpler methods, such as support vector machines, work better with this dataset?
- The machine learning model that gives the best performance is further studied by WEKA attribute evaluation. Will the same features be identified in other machine learning models with the same WEKA analysis? If so, this could also support the use of AI methods.
Another minor point:
Some markers in Fig. 2 are not clearly explained. For instance , c21 of group an is indicated by a red circle. I'm not sure if this is the case because it contrasts with other group members' activities. If there is just one outlier in this group that has low activity, I think it also shows non-homogeneous deterioration activity.
Author Response
In order to extract information that can inform the design of PROTACs, the authors describe a thorough study of the androgen receptor degradation data in this paper. The study covers a broad range of the properties of the degraders or the complex they form with proteins. The work is solid, and the authors make clear arguments regarding the choice of methods and provide proper discussions on the results.
Although it seems sense to use machine learning approaches in this study because they have proven to be effective at sifting out key features from challenging datasets, the choice of machine learning method needs more justification. Because the number of dataset is relatively small in this study, such a limitation could lead to the failure of many machine learning methods. With this concern, I would suggest the following point might need to be addressed or discussed:
- Could the author elaborate more on their decision-making process for ML methods? How can they ensure that the overfitting issue won't arise?
Good point. At first, we performed a multilinear regression (not shown), however we noted that the models obtained were not well-performing. Thus, we opted for the implementation of several classification models aiming at discriminating active from inactive compounds. To avoid overfitting, we divided the overall dataset into the training and the validation test set with a ratio of 80:20. We performed a 10-fold cross-validation to train the model to reduce the bias and increase the variance. ROC and MCC values are similar between the training and the test set suggesting that overfitting is not an issue. This seems to be confirmed by the Y-randomization that was used to validate the models implemented. Moreover, considering the high correlation present between several molecular descriptors we implemented 3-descriptors models (Table). Random Forest 3-descriptors model further confirms the trend identified with the 7-descriptors model.
3-descriptors models (nHAcc, TPSA, MW)
Model |
Sensitivity |
Specificity |
MCC |
Roc Area |
|
VHL |
RF training |
0.737 |
0.870 |
0.615 |
0.839 |
RF test |
0.800 |
0.833 |
0.633 |
0.767 |
|
CRBN |
RF training |
0.500 |
0.538 |
0.038 |
0.603 |
RF test |
0.800 |
1.000 |
0.775 |
0.967 |
|
Entire dataset |
RF training |
0.594 |
0.619 |
0.211 |
0.711 |
RF test |
0.875 |
0.800 |
0.671 |
0.888 |
- Will simpler methods, such as support vector machines, work better with this dataset?
Good suggestion. We tested SVM as implemented in Orange, but we obtained models with slightly lower quality than Random Forest. We added a comment in the text.
- The machine learning model that gives the best performance is further studied by WEKA attribute evaluation. Will the same features be identified in other machine learning models with the same WEKA analysis? If so, this could also support the use of AI methods.
Machine learning is an AI-based approach that we applied to extract information from a dataset of published AR PROTACs. The aim of the study is to help medicinal chemists to get the most from available datasets by in silico resources widely applied in drug discovery, easy to use and not requiring high level of expertise. The AI methods suggested by the Reviewer (we guess Deep Learning techniques) are beyond the skills of medicinal chemists and although of potential interest are right now not ready to be implemented at this early stage of drug discovery.
Another minor point:
- Some markers in Fig. 2 are not clearly explained. For instance, c21 of group a-n is indicated by a red circle. I'm not sure if this is the case because it contrasts with other group members' activities. If there is just one outlier in this group that has low activity, I think it also shows non-homogeneous deterioration activity.
The comment is sound. Since we clustered together compounds both sharing the same scaffold and having different scaffold with a single localized structural feature (i.e., ring system substitution) cluster A includes 8 compounds characterized by two main scaffolds. c10, and c16-c20 degraders show the same Murcko Scaffold, meanwhile, compound c21 presents the substitution of the 3-methylisoxazole group with an acetyl group, determining a variation in the ring systems included in the framework structure. This explains the anomalous position of c21 in Fig. 2. The text was clarified and revised accordingly in paragraph 2.3.
Reviewer 2 Report
After evaluating the manuscript “IIn silico tools to extract the drug design information content of degradation data: the case of PROTACs targeting the Androgen Receptor" I have to recommend its major revision in current version
1. All graphs, figures and tables must be of proper quality. Image resolution must be at least 600 dpi.
2. Unfortunately, the authors of the work do not discuss the role of the linker in chimeric molecules. The sample of studied ligands contained two types of linkers, conformationally rigid and conformationally flexible. The typology and conformational dynamics of the linker are the key points in the formation of the ternary complex and, as a consequence, the efficiency of POI degradation. This issue should be discussed.
3. The authors note that at the moment there is a heterogeneity of experimental data on the in vitro assessment of degradation efficiency, which hinders the quality of SAR. However, at the moment there are many reporter lines that allow you to effectively assess the degradation efficiency under standardized conditions and, as a result, draw conclusions about SAR. (See for example HIBIT cell lines from Promega). The discussion about this issue should be revised in the manuscript.
4. In the introduction of the manuscript, it is necessary to consider other molecular modeling approaches used to evaluate future and described PROTAC.
Author Response
After evaluating the manuscript “In silico tools to extract the drug design information content of degradation data: the case of PROTACs targeting the Androgen Receptor" I have to recommend its major revision in current version
- All graphs, figures and tables must be of proper quality. Image resolution must be at least 600 dpi.
OK, done
- Unfortunately, the authors of the work do not discuss the role of the linker in chimeric molecules. The sample of studied ligands contained two types of linkers, conformationally rigid and conformationally flexible. The typology and conformational dynamics of the linker are the key points in the formation of the ternary complex and, as a consequence, the efficiency of POI degradation. This issue should be discussed.
Good point. A new section focusing on the linkers has now been introduced.
- The authors note that at the moment there is a heterogeneity of experimental data on the in vitro assessment of degradation efficiency, which hinders the quality of SAR. However, at the moment there are many reporter lines that allow you to effectively assess the degradation efficiency under standardized conditions and, as a result, draw conclusions about SAR. (See for example HIBIT cell lines from Promega). The discussion about this issue should be revised in the manuscript.
We agree with the Reviewer and the text, and the Conclusions were modified accordingly.
- In the introduction of the manuscript, it is necessary to consider other molecular modeling approaches used to evaluate future and described PROTAC.
Ok. We changed the text accordingly and we explained the rationale for applying the tools described in the text.
Round 2
Reviewer 2 Report
The authors answered all my questions to the full. I recommend that you accept the article in its present form.